# Autoimmune Thyroid Diseases and Physical Activity and Sports—More Unknowns than Facts

**DOI:** 10.3390/biomedicines13102352

**Published:** 2025-09-25

**Authors:** Monika Skrzypiec-Spring, Andrzej Pokrywka, Adam Szeląg, Agnieszka Zembroń-Łacny

**Affiliations:** 1Department of Pharmacology, Wroclaw Medical University, 50-345 Wroclaw, Poland; adam.szelag@umw.edu.pl; 2Department of Biochemistry and Pharmacogenomics, Medical University of Warsaw, 02-097 Warsaw, Poland; andrzej.pokrywka@wum.edu.pl; 3Faculty of Medicine, University of Warsaw, 02-089 Warsaw, Poland; a.zembron-lacny@cm.uz.zgora.pl

**Keywords:** autoimmune thyroid disease, Hashimoto’s thyroiditis, Graves’ disease, physical activity, hypothyroidism, hyperthyroidism

## Abstract

**Background:** Chronic autoimmune thyroiditis, also known as Hashimoto’s thyroiditis (HT) or chronic lymphocytic thyroiditis whose pathophysiology includes both cellular (T-cell mediated) and humoral (B-cell mediated) immune responses, leads to the destruction of thyroid follicular cells and progressive fibrosis of the thyroid gland. While hypothyroidism is a common autoimmune disease, athletes may experience unique challenges related to its diagnosis and management within the context of training programme, competition and anti-doping regulations. In turn, it is known that moderate physical exercise can have a positive effect on the immune system, while excessive exercise can cause unfavourable changes in this system. Therefore, we aimed (1) to identify the interplay between physical activity and autoimmune thyroid disease, (2) to quantify changes in thyroid function associated with physical activity, and (3) to explain the underlying mechanisms of autoimmune thyroiditis in athletes. **Methods:** The medical database PubMed/MEDLINE was searched in the time period 2004–2025, where 12 publications met the inclusion criteria and were ultimately included for further evaluation according to the RAMESES (Realist and Meta-narrative Evidence Syntheses: Evolving Standards). **Results:** The reviewed studies have clearly indicated that physical exercise has a beneficial effect on thyroid function, and two studies reported that non-excessive physical exercise leads to a decrease in TPO-Ab concentrations. **Conclusions:** The beneficial effect of physical exercise on thyroid function and immune response underlines the need for further well-designed studies to formulate specific guidelines for patients with HT, as well as for athletes with autoimmune thyroid disease. Similarly, there is a need to evaluate the prevalence of thyroid hormone use among amateur and professional athletes in order to establish prevention strategies.

## 1. Introduction

Chronic autoimmune thyroiditis, also known as chronic lymphocytic thyroiditis, was first described by Japanese physician Haruto Hashimoto in 1912 [1]. Along with Graves’ disease (GD), the condition falls into the category of autoimmune thyroid disorders [2]. In recognition of Hashimoto’s contribution, the disease is commonly referred to as Hashimoto’s thyroiditis (HT). In developed countries, HT is the most common cause of hypothyroidism [3], particularly in iodine-sufficient areas [4]. A systematic review and meta-analysis of studies from multiple countries estimated the global prevalence of HT at 7.5%, with a higher prevalence of 11.4% reported in low- and middle-income areas [5]. The prevalence in women is four times higher than in men [5]. Limited data are available on the prevalence of HT in athletes; however, the prevalence of subclinical hypothyroidism in competitive athletes has been reported to be 9.5% [6].

Hashimoto’s thyroiditis is an autoimmune disease whose pathophysiology includes both cellular (T-cell mediated) and humoral (B-cell mediated) immune responses, leading to the destruction of thyroid follicular cells, and progressive fibrosis of the thyroid gland [7]. The cellular immune response involves both CD8+ and CD4+ T cells. CD8+ cytotoxic cells play a key role in the immune dysfunction underlying HT by infiltrating the thyroid tissue, causing inflammation and destruction of follicular cells [8]. CD4+ T cells, including Th1, Th2, Th17, and regulatory T cells (Tregs), contribute to disease pathogenesis through distinct mechanisms [9]. Th1 cells are upregulated in HT and promote inflammation in thyroid tissue. Conversely, Tregs, which are essential in maintaining immune tolerance, are downregulated in HT. This loss of immune regulation facilitates the activation of other immune cells, especially macrophages and B cells, leading to the production of the following polyclonal antibodies: anti-thyroid peroxidase antibody (TPO-Ab), anti-thyroglobulin antibody (TG-Ab) and thyroid-stimulating hormone (TSH) receptor antibody (TSHR-Ab) [9,10]. TPO-Ab is the most prevalent antibody in HT, present in over 90% of affected individuals [11], while TG-Ab is detected in 50% to 80% of cases. Multiple types of TSHR-Abs have been identified, including stimulating, blocking, or neutral antibodies [12]. In HT, blocking antibodies are the most common type. However, in some HT cases, stimulating TSHR-Abs are present, which may explain the progression of HT to GD or the development of thyroid-associated orbitopathy [13]. The clinical presentation of HT is variable and includes euthyroid state, subclinical hypothyroidism and overt hypothyroidism, which are the most frequent presentations, or less commonly, hyperthyroidism, also referred to as “hashitoxicosis”, thyroiditis and postpartum thyroiditis, thyroid nodularity and ocular involvement [14]. Although the majority of individuals with HT remain euthyroid, approximately 20% to 30% eventually develop hypothyroidism [15].

The GD is the most common cause of hyperthyroidism in regions where iodine deficiency does not occur [16]. Although GD can affect individuals of any age and sex, it most frequently occurs in women between the fourth and fifth decades of life [16]. The pathogenesis of GD involves persistent activation of thyroid-stimulating hormone receptor (TSHR) on thyroid follicular cells by TSHR-Ab, leading to hyperthyroidism and often thyroid enlargement, clinically manifested as a goitre [17]. The pathophysiology of this autoimmune disease involves multiple regulatory mechanisms including regulatory T cell failure, proliferation of autoreactive T and B cells, increased Human Leukocyte Antigen D affinity for the TSH receptor, leading to more efficient antigen presentation, or the presence of a more immunogenic TSH receptor haplotype or increased exposure to the TSH receptor peptide [18,19].

Both the deficiency of thyroid hormone in target tissues in hypothyroidism and their excess in hyperthyroidism can vary in severity. Given the significant effects on skeletal muscles’ function and the cardiovascular system, these hormonal imbalances may potentially affect exercise capacity and physical performance. Exercise capacity is substantially and directly impacted by thyroid hormones’ influence on endurance, muscle strength and post-exercise regeneration. The positive effect of thyroid hormones on endurance is complex. They exert a permissive effect on catecholamines, increasing the expression of beta receptors, which results in increased heart rate, stroke volume, cardiac efficiency, and contractility [20]. In the respiratory system, they increase oxygenation [20]. Thyroid hormones stimulate mitochondriogenesis and thereby augment cellular oxidative capacity [21]. They stimulate mitochondrial oxidative phosphorylation, which leads to ATP production [21]. They also improve endurance by regulating enzymes involved in both lipolysis and lipogenesis as well as insulin-dependent glucose uptake, and both gluconeogenesis and glycogenolysis, leading to increased supply of energy substrates during prolonged physical exercise [22,23]. Thyroid hormone deficiency has the opposite effect on all of the above-described processes and leads to reduced endurance. Thyroid hormones also produce a complex effect on skeletal muscles. They regulate the growth and differentiation of fast-twitch type II muscle fibres that are necessary for powerful movements [23]. The impact of thyroid hormones on skeletal muscle properties includes: increased rate of contraction and relaxation, decreased energetic efficiency of contraction due to higher ATP consumption at rest and during activity, increased glycolytic capacity and increased mitochondrial density which both lead to enhanced ATP generation [24]. Therefore, thyroid hormone deficiency may impair muscle strength, contraction and relaxation. Moreover, thyroid hormones support muscle remodelling and repair after intense exercise. They help reduce muscle discomfort and accelerate recovery by stimulating protein synthesis and improving the turnover of damaged proteins [25]. In hypothyroidism associated with Hashimoto’s disease, the post-exercise regeneration process is weakened.

On the other hand, non-excessive physical activity has been shown to significantly increase the number of T-regulatory cells and shift the Th1/Th2 balance towards reduced Th1 cell production and decreased immunoglobulin secretion [26,27]. Furthermore, it has been proven that physical activity stimulates the release of interleukin-6 from skeletal muscles, which subsequently induces an anti-inflammatory response via interleukin-10 secretion and interleukin-1β inhibition [26,27]. These immunomodulatory effects of physical activity on the immune system may have a potentially beneficial impact on the immunological processes underlying autoimmune thyroiditis. On the other hand, intense physical exercise leads to a transient immune dysfunction. Specifically, for several hours or days after prolonged and intense physical exercise, T-cell, NK, and neutrophil function is impaired, the balance of type I and II cytokines is disturbed, and the expression of major histocompatibility complex II in macrophages is altered. Additionally, immune responses to primary and recall antigens are weakened and delayed [28].

Although pathophysiological mechanisms suggest impaired exercise capacity in both HD and GD, and there is interest in the beneficial benefits of moderate physical activity on the immunological processes underlying these diseases, there are, to the authors’ knowledge, no clear recommendations regarding activity and sports participation in individuals suffering from them. Therefore, there is a need to identify the type and intensity of physical activity which can be recommended to patients with autoimmune thyroid disease. The high percentage of athletes affected by this condition underscores the importance of developing evidence-based guidelines for training and treatment, both during the chronic phase of the disease and during exacerbations. To this end, it is essential to determine which types and intensities of physical activity have a proven impact on the course of autoimmune thyroid disease. In our study, we aimed to conduct a comprehensive scientific review of the literature published between 1 May 2004 and 1 May 2025 regarding the relationship between physical activity and sports and autoimmune thyroid diseases. We sought to identify the types and intensities of physical activity that show the strongest evidence of a beneficial effect on disease progression, explore the mechanisms involved, and evaluate research on the impact of physical activity and sports on laboratory test results. Additionally, we aimed to highlight the influence of medications used in the treatment of HT on physical performance and to consider their potential misuse as doping agents.

## 2. Materials and Methods

This review was conducted in accordance with RAMESES (Realist and Meta-narrative Evidence Syntheses: Evolving Standards) guidelines, following the principles of a realist review [29,30]. This method was selected because it is a theoretically and interpretively based literature review that seeks to answer the question of whether a given intervention works, for whom, under what circumstances, and how. It enables the development of basic theories for an intervention programme and subsequent analysis of the existing evidence to assess their validity and practical applicability. The aim of this review was to identify the type and intensity of physical activity with the most well-documented influence on autoimmune thyroid disease, to quantify changes in thyroid function associated with physical activity, to explore the underlying mechanisms and influencing factors and present theory for intervention programme aimed to formulate recommendations.

Conceptual model diagrams illustrating key mechanisms or hypothetical pathways are presented in Figure 1.

Following the principles of a realist review, the process began with an initial search of the literature (Figure 2 and Figure 3) [29,30]. Search terms included: “physical activity”, “sports”, “autoimmune thyroid disease”, “autoimmune thyroiditis”, “Hashimoto’s thyroiditis”, “Graves’s disease”, “hypothyroidism”, “hyperthyroidism”, and “thyrotoxicosis”. The PubMed/MEDLINE database was searched for peer-reviewed articles published between 1 May 2004 and 1 May 2025, limited to studies in English. Additionally, a separate search was performed using the terms: “thyroid hormones”, “levothyroxine, “triiodothyronine” and “doping” over the same time period. The final search was conducted on 1 May 2025. Titles and abstracts were screened by reviewers based on methodological criteria (e.g., including only clinical trials, crossover studies, systematic reviews) and relevance to the topic. Each article was assessed independently by at least two reviewers at each stage of the selection process. Studies meeting the inclusion criteria were selected and reviewed in full to further discuss potential relationship between physical activity and sport and autoimmune thyroid disease. Data from each article were then extracted by two reviewers from the research team. The quality of empirical studies was assessed using the Mixed Method Appraisal Tool (MMAT) which provides a rationale to evaluate the strengths of evidence based on the contribution to model development [24]. Discrepancies in MMAT ratings were resolved through discussion until a consensus was reached. Although the initial search identified a substantial number of studies, only a few directly investigated the potential relationship between physical activity and sport and autoimmune thyroid disease. Using the identified literature, the research team developed a preliminary conceptual model. A second search was then conducted using PubMed/MEDLINE and Google Scholar to validate and expand the initial findings. This resulted in two additional studies which were assessed using the same data extraction and quality appraisal procedures described above. These studies were subsequently included in the final review.

## 3. Results

The final analysis included 12 publications: 7 studies investigated the effects of autoimmune thyroid disease on physical activity and sport, while 5 studies examined the impact of physical activity and sport on the course of autoimmune thyroid disease. No research papers were found on the use of thyroid hormones as doping except case studies and review articles. Table 1 summarizes main information from the scientific studies on the effects of autoimmune thyroid disease on physical activity and sport, including study design, participant characteristics, and main findings. Table 2 presents the scientific studies exploring the effects of physical activity and sport on the progression of autoimmune thyroid disease, along with main outcomes and conclusions.

The main information from the tables is also summarized in the form of evidence gap maps of findings (Figure 4 and Figure 5).

## 4. Discussion

### 4.1. The Effects of Autoimmune Thyroid Disease on Physical Activity and Sport

The pathophysiology of autoimmune thyroid disease, both HT and GD, indicates that the course of both conditions may be associated with changes in the cardiovascular, musculoskeletal, and metabolic systems, which may impair physical capacity and reduce the ability to perform physical activity. This is confirmed by the results of studies on the effect of hypothyroidism and hyperthyroidism on exercise capacity and the ability to perform exercises, although such studies are limited in number, to a few or even a single study, often involving small cohorts and/or focusing on observational data—particularly in the case of GD. While the symptoms of HT are typically expected in overt, untreated, hypothyroidism, Tanriverdi et al. demonstrated that even patients with subclinical hypothyroidism were less physically active than healthy control women and exhibited more muscular-neural symptoms and decreased strength [35]. Similarly, Hanke et al. showed that levothyroxine treatment in subclinical hypothyroidism led to improvements in strength, mobility, and endurance [33]. All studies involving patients with overt, untreated hypothyroidism consistently report reduced exercise tolerance [36,37]. However, there are conflicting data regarding the occurrence of impaired exercise tolerance in people treated with levothyroxine. For example, Mainenti et al. observed improved submaximal cardiopulmonary exercise capacity six months after TSH normalization [37], while Lankhaar et al. reported both an improvement in exercise tolerance with treatment, and persistent exercise intolerance in some patients despite treatment [36]. In another study conducted by the same research centre, two-thirds of patients treated with levothyroxine reported that hypothyroidism limited their physical activity, which was more pronounced in patients with autoimmune thyroiditis [34]. Conversely, Tian in the American population and Gacek in the Polish population demonstrated that patients with autoimmune thyroid disease maintained an adequate level of physical activity [31,38]. Only one study meeting the inclusion criteria concerned the effect of hyperthyroidism on tolerance and ability to undertake exercise. As expected from the disease’s pathophysiology, it showed that patients with hyperthyroidism experienced reduced exercise capacity, decreased respiratory muscle strength and endurance, lower physical activity levels and diminished quality of life [32]. While the reduction in exercise tolerance in overt untreated hypothyroidism or overt hyperthyroidism is consistent with the underlying pathophysiology, the inconsistency of the study results of reduced exercise tolerance in patients with normal thyroid hormone levels in subclinical hypothyroidism and in patients successfully treated with levothyroxine is intriguing, because the impaired exercise tolerance recorded in normal thyroid concentrations is difficult to be explained pathophysiologically. The discrepancies may result from small sample sizes and failure to control for other factors affecting exercise capacity. Moreover, most studies relied on patients’ subjective assessment of exercise tolerance and only in one study, involving a small group of 23 participants, was the submaximal cardiopulmonary exercise performance objectively assessed. Furthermore, the objective assessment of the amount of physical activity in patients with autoimmune thyroiditis and HT treated with levothyroxine, demonstrating sufficient activity, suggests that the subjective methods of physical activity assessment such as questionnaires may be inadequate. This emphasizes the need for objective methods to assess exercise tolerance in patients with subclinical hypothyroidism and patients with overt hypothyroidism treated with levothyroxine.

### 4.2. The Effects of Physical Activity and Sport on the Course of Autoimmune Thyroid Disease

Although the beneficial effects of physical activity on the autoimmune system are well established, there are no clear recommendations regarding the amount and type of physical activity suitable for the patients with autoimmune thyroid disease. This may be due to the lack of sufficient and reliable data assessing the effect of physical exercise on thyroid function in this patient population. The gap in the available literature is particularly surprising given the high and growing incidence of these diseases, especially HT. Only five publications on this topic met the search criteria. A systematic review by Duñabeitia et al. [42] showed a trend towards reduced thyroid-stimulating hormone levels with physical activity, although it was not statistically significant. Almas et al. [40], however, demonstrated significant improvement in heart rate on-kinetics and physical activity level in patients with subclinical hypothyroidism following 12 weeks of endurance training. However, this study was conducted on a small sample. Similar results were obtained in a larger cohort by Vuletić et al. [39] who showed that in patients with hypothyroidism, recreational physical activity led to a reduction in TSH and TPO-Ab levels. In contrast to recreational exercise, occupational physical activity, defined as mandatory activity performed during the standard eight-hour workday, often with reduced opportunities for rest, correlated with decreased thyroid function and increased thyroid autoimmunity, which was particularly pronounced in patients with overt, untreated hypothyroidism. This is consistent with the research results of Tian et al., who analysed data from 5877 American adults in The National Health and Nutrition Examination Survey 2007–2012. They found that thyroid function was significantly affected by weekly physical activity volume and duration, but the relationship between physical activity and thyroid disease was non-linear [38]. It is noteworthy that Tian et al. reported that physical activity, expressed as the metabolic equivalent of task (MET)-minutes/week and total physical activity time, were not significantly associated with thyroid function in women. This finding also applied to the total population and to men. However, in the total population, higher levels of weekly physical activity and total physical activity time were generally associated with a decreasing trend in TSH, free thyroxin (FT4), and total thyroxin T4 levels and an overall increasing trend in free triiodothyronine (FT3) and total triiodothyronine (T3) levels. TPO-Ab showed a downward trend, however; it should be added that after excluding participants who tested positive for antibodies, reanalysis revealed no obvious changes in the results. The downward trend in TPO-Ab concentration with increasing physical activity, up to approximately 5000 MET-minute/week, when individuals with autoimmune thyroiditis were included in the analysis, may indicate a beneficial effect of this level of physical activity on autoimmune processes. The International Physical Activity Questionnaire allowed us to assess the level of total physical activity in four categories: vigorous activity (above 1500–3000 MET-min/week), moderate activity (600–1500, 600–3000 MET-min/week), walking (below 600 MET-min/week) and sitting. Among these, vigorous physical activity was associated with the most pronounced beneficial effect on TPO-Ab levels [43]. Matsumura et al. [41] analysed a group of 1222 female non-elite runners aged ≥ 35 years and found that training intensity and duration, including average miles per week, training pace, or years of accumulated running, did not differ between female runners reporting hypothyroidism and those who did. These outcomes suggest a lack of association between training intensity and the diagnosis of hypothyroidism. It should be noted that all runners were nor not professional athletes and average weekly millage for most of them ranged from 11 to 40 miles/week, which corresponds to the recommendations for recreational runners, therefore the reported training volumes cannot be considered excessive [44]. The data from these few studies are insufficient to formulate objective conclusions and recommendations. Nevertheless, in all the studies presented, non-excessive physical exercise was shown to have a beneficial effect on thyroid function. Only two studies specifically assessed the effect of physical exercise on the immune system function; in both, non-excessive physical exercise led to a decrease in TPO-Ab concentrations. These findings suggest that physical exercise may positively influence both immunological processes and thyroid function. The results highlight the need to conduct well-designed studies to establish specific guidelines regarding the recommended types and levels of physical exercise for patients with hypothyroidism and HT. Professional athletes suffering from hypothyroidism or HT represent a distinct clinical group. In cases of excessive exertion and relative energy deficiency, they may develop thyroid dysfunction resulting from functional disturbances of the pituitary gland [45]. To our knowledge, there are no available studies that assess the frequency of this phenomenon and describe the relationship between pituitary dysfunction in the thyrotropic axis and the intensity of physical exertion. Therefore, there is also a clear need to conduct well-designed studies and to develop evidence-based recommendations.

To our knowledge, there are no publications assessing the effect of physical activity on the course of GD, nor are there recommendations regarding the timing of return to physical activity or professional sports in these patients. For this reason, there is a need to conduct research on how resuming physical activity affects the course of GD and the immune system, as well as studies determining the optimal timing and intensity of physical activity in this population.

### 4.3. Thyroid Hormones as Potential Doping

Although thyroid hormones are not classified as doping agents by the World Anti-Doping Agency (WADA), there are theoretical grounds that may explain the possible influence of thyroid hormones on the ability to improve performance. They increase the expression of beta receptors, exerting a permissive effect on catecholamines [20]. They stimulate mitochondriogenesis, which results in increased cellular oxidative capacity and mitochondrial oxidative phosphorylation, increasing ATP production [21]. They also regulate enzymes involved in lipolysis, lipogenesis, insulin-dependent glucose uptake, and both gluconeogenesis and glycogenolysis, thereby increasing the supply of energy substrates [23,24]. They regulate the growth and differentiation of fast-twitch type II muscle fibres, increase the rate of contraction and relaxation, reduce the energy efficiency of contraction due to higher ATP consumption at rest and during activity, increase glycolytic capacity with subsequent ATP generation, and increase mitochondrial density, leading to increased ATP generation [24,25]. Exogenously administered TH are therefore attractive to professional athletes and bodybuilders who want to achieve an appropriate weight and low body fat. Additionally, these drugs are illegally used to counteract changes in thyroid hormone levels that can occur during the use of anabolic-androgenic steroids [28]. The extent of thyroid hormones use can be inferred from studies assessing the prevalence of anabolic-androgenic steroid use. Among these, the publication by Skrzypiec-Spring et al. (2024) [46] provides data on thyroid hormones use. The authors reported that 35.42% of individuals who used testosterone illegally and exercised in municipal gyms in Wroclaw, Poland, also used thyroid hormones. Testosterone, in turn, was used illegally by 35% of the surveyed amateur athletes. Although thyroid hormones were used by only about 12% of those who trained, the potential for serious complications resulting from their use may pose a serious toxicological problem. To the authors’ knowledge, there are no available studies summarizing the toxicological significance of illegal thyroid hormone use; however, numerous clinical case reports document instances of poisoning in bodybuilding athletes, including a fatal case [47,48,49,50,51,52,53,54,55,56]. From a clinical management perspective, it is essential to consider the possibility of illicit drug use in individuals practicing sports such as bodybuilding when analysing laboratory tests and clinical presentations. Early detection is critical to avoid misdiagnosis or delayed treatment [57]. Furthermore, both primary and secondary prevention strategies should also be implemented to mitigate thyroid hormones abuse.

Illegal use of thyroid hormones by professional athletes was also recorded by Handelsman et al. [58]. The authors measured total thyroxine (T4), T3 and reverse T3, TSH, free thyroxine (FT4) and FT3 concentrations in 498 frozen serum samples collected from Australian athletes during doping control tests. In addition, they examined the athletes’ mandatory declarations on Doping Control Forms which listed all medications used in the week prior to testing. Two athletes were diagnosed with biochemical thyrotoxicosis (incidence 4/1000 athletes). Similarly, only 2 athletes declared levothyroxine use and none of them declared triiodothyronine use, yielding the same incidence rate of 4/1000 athletes. Due to data protection regulations, it was not possible to determine whether thyrotoxicosis occurred in individuals who had declared levothyroxine use.

The prevalence of thyroid hormones use among elite athletes who participated in PyeongChang 2018 Winter Olympics Games, Minsk 2019 European Games, Tokyo 2020 Olympics and Paralympics Games, based on self-disclosure in Doping Control Form, was estimated at 1.4%, which is slightly higher than most reported rates of thyroid hormones use in the general community [47]. For a substance or method to be included in the WADA prohibited list, it must meet at least two of the following three criteria: (1) it has the potential to enhance or enhances sport performance; (2) it poses an actual or potential health risk to the athlete; (3) it violates the spirit of sport [59]. There is evident room for interpretation in the definition of ‘the spirit of sport’, since scientific evidence is sometimes lacking to establish a causal link between the use of a given substance and sport performance [60]. According to Gild et al. [47], thyroid hormones currently do not meet 2 of the 3 criteria required by the WADA Code for inclusion on the Prohibited List. This conclusion also takes into account considerations related to feasibility, logistics, and burden on the athlete. On balance, therefore, prohibition of TH in elite sport under the Code is not at present justified. However, more convincing evidence on the prevalence of TH use among elite athletes and, in particular, whether such use has any performance-enhancing effects, might shift this assessment in the future [60]. From a sports medicine perspective, the inclusion of thyroid hormones on the list of prohibited substances in sport will necessitate the use of Therapeutic Use Exemption (TUE), which means that athletes would have to obtain WADA approval to use the medication for medically justified reasons. Given the prevalence of Hashimoto’s thyroiditis, which requires thyroid hormone treatment, such a policy change this could significantly increase the workload on the regulatory bodies responsible for evaluating TUE applications and may complicate the approval process. Importantly, without a valid TUE, athletes undergoing thyroid hormone therapy would be ineligible to participate in competitive sports. The problem of autoimmune thyroid disease and physical activity and sports is complex and not yet fully understood. In our study, we decided to use realist synthesis not only to explore this relationship but also to identify the types and intensities of physical activity that have the strongest evidence of a positive effect on the disease progression. Additionally, we aimed to investigate the underlying mechanisms responsible for these effects. Although the authors consider realist synthesis the best method to achieve this study objectives, the choice of the method also presents certain limitations. Specifically, while standards exist for reporting realistic synthesis, there are no established guidelines for conducting such studies or developing protocols. This lack of standardization allows for methodological flexibility and inclusiveness but also creates the risk of suboptimal data analysis. Another limitation is that the analysed publications did not provide sufficient evidence to determine whether factors such as gender, age, or disease duration modified the effect of exercise on the immune system and thyroid function in autoimmune disease. Similarly, only a limited number of studies accounted for treatment as a potential confounding factor. While some findings suggested that the impact of exercise on thyroid function may be more pronounced in treated patients, the available data are insufficient to support definitive conclusions. Nevertheless, as realistic synthesis facilitates understanding of why and how a factor works, supports informed decision-making, and enables to formulate recommendations, it remains the only method demonstrated to effectively meet the aims of this study, despite its limitations.

## 5. Conclusions

Although autoimmune thyroiditis, and especially HT, is highly prevalent, there is limited data on how the disease affects physical performance and on the impact of physical activity and sports on its progression. Based on current evidence, it is not possible to formulate specific recommendations regarding the type and extent of physical activity for patients with autoimmune thyroid disease, nor to provide clear guidelines regarding sports participation. Nevertheless, all the reviewed studies indicated that non-excessive physical exercise had a beneficial effect on thyroid function and on immunological processes. Only 2 studies assessed the effect of physical exercise on the function of the immune system and both studies reported that non-excessive physical exercise led to a decrease in TPO-Ab concentrations. The beneficial effect of physical exercise on immunological processes and thyroid function demonstrated in these studies underlines the need for well-designed studies to formulate specific guidelines for patients with hypothyroidism and HT, as well as for athletes with autoimmune thyroid disease. Similarly, there is a need to evaluate the prevalence of thyroid hormone use among amateur and professional athletes in order to establish prevention strategies.

## Figures and Tables

**Figure 1 biomedicines-13-02352-f001:**
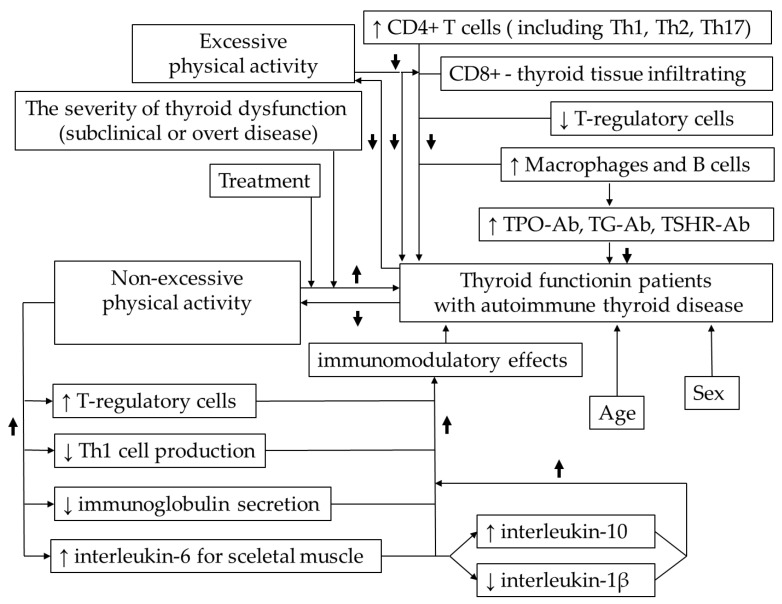
Conceptual model diagram illustrating key mechanisms or hypothetical pathways of the interdependencies between physical activity, thyroid function and immune system. ↑—increase, ↓—decrease.

**Figure 2 biomedicines-13-02352-f002:**
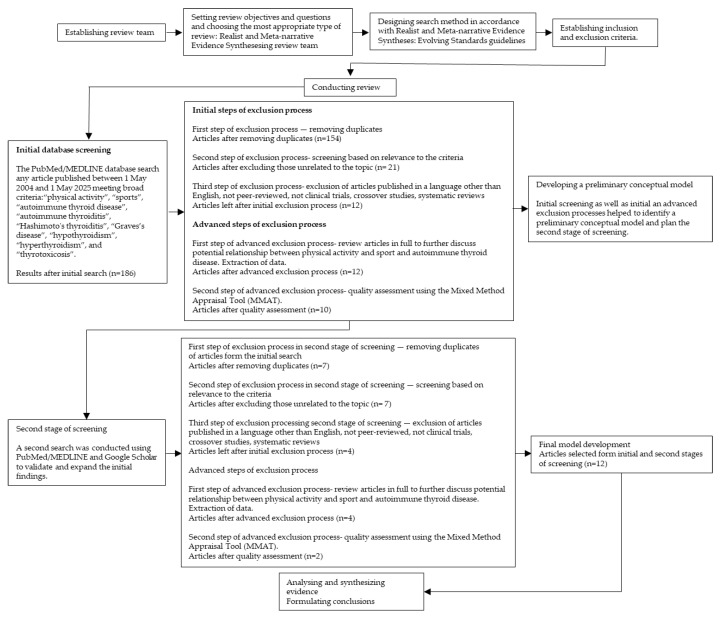
Study selection flow chart for realist and meta-narrative evidence synthesis based on Realist and Meta-narrative Evidence Syntheses: Evolving Standards guidelines by Pawson R et al. [29]. Search terms: “physical activity”, “sports”, “autoimmune thyroid disease”, “autoimmune thyroiditis”, “Hashimoto’s thyroiditis”, “Graves’s disease”, “hypothyroidism”, “hyperthyroidism”, and “thyrotoxicosis”.

**Figure 3 biomedicines-13-02352-f003:**
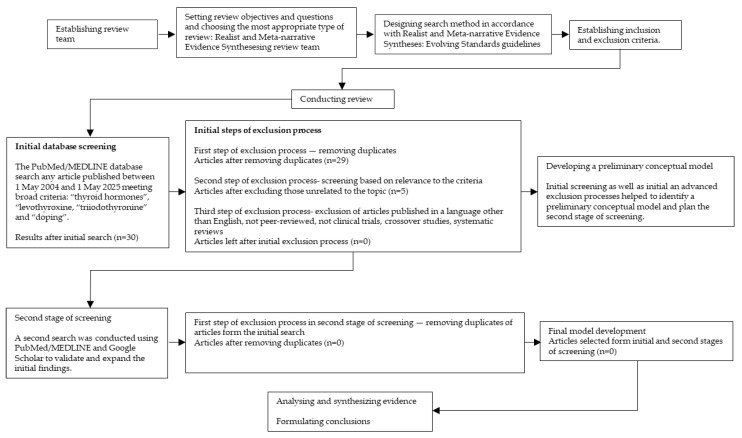
Study selection flow chart for realist and meta-narrative evidence synthesis based on Realist and Meta-narrative Evidence Syntheses: Evolving Standards guidelines by Pawson R et al. [29]. Search terms: “thyroid hormones”, “levothyroxine, “triiodothyronine” and “doping”.

**Figure 4 biomedicines-13-02352-f004:**
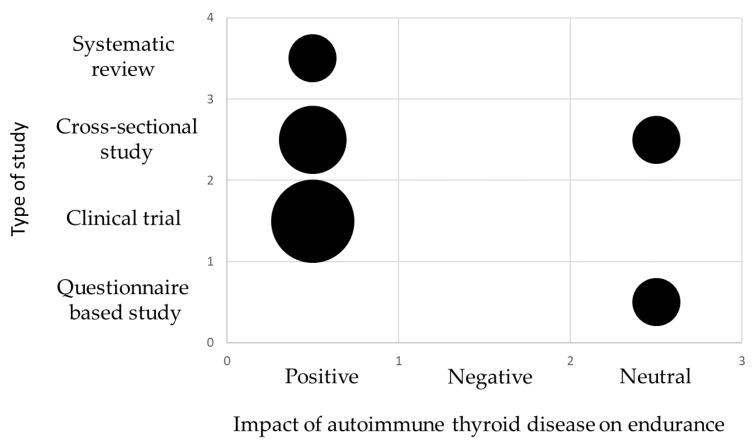
Evidence gap map showing main information about the scientific studies and the key findings on the impact of autoimmune thyroid disease on endurance. The size of each circle is proportional to the number of articles. The smallest circle represents 1 publication.

**Figure 5 biomedicines-13-02352-f005:**
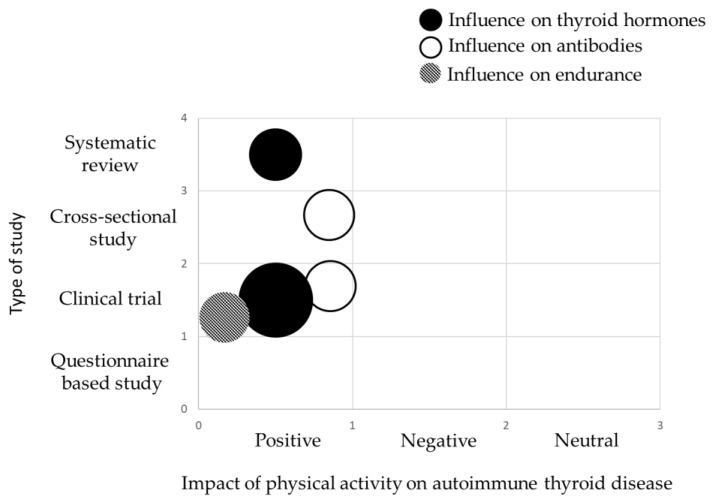
Evidence gap map showing main information about the scientific studies and the key findings on the effects of physical activity and sport on the course of autoimmune thyroid disease. Each colour represents what was targeted by physical activity, while the size of each circle is proportional to the number of articles. The smallest circle represents 1 publication.

**Table 1 biomedicines-13-02352-t001:** Main information about the scientific study and the key findings on the effects of autoimmune thyroid disease on physical activity and sport.

Publication	Type of Publication	Main Findings	Study Group
Gacek et al., (2025) [31]	Questionnaire based study	Among women with Hashimoto’s disease treated with levothyroxine the physical activity was average and sufficient. Moreover, moderate physical activity was associated with a lower intensity of depressive symptoms.	219 women with Hashimoto’s disease treated with levothyroxine (mean age 33.8 ± 9.9)
Yılmaz et al., (2024) [32]	Clinical trial	Exercise capacity, respiratory muscle strength and endurance, physical activity level, dyspnoea, and quality of life were affected in patients with hyperthyroidism	16 patients with hyperthyroidism and healthy controls.
Hanke et al., (2023) [33]	Clinical trial	Treatment with levothyroxine in subclinical hypothyroidism improves strength, mobility and endurance performance	25 women (mean age 27.36 ± 5.77) with subclinical hypothyroidism
Lankhaar et al., (2021) [34]	Cross-sectional matched case–control study	Two-thirds of patients reported limited physical activity performance. This was more pronounced in patients with autoimmune thyroiditis	1724 women (mean age 53.0 years ± 11.6) and 1802 controls (mean age 52.6 ± 13.2) treated with lewothyroxine
Tanriverdi et al., (2019) [35]	Cross-sectional study	Women with subclinical hypthyroidism had lower physical activity level compared to healthy controls	32 women with newly diagnosed subclinical hypthyroidism and 28 healthy women
Lankhaar et al., (2014) [36]	Systematic review	Exercise intolerance is observed in untreated patients with hypothyroidism. In some patient’s persistent exercise intolerance is recorded, despite treatment.	38 studies, 1379 patients with hypothyroidism
Mainenti et al., (2009) [37]	Clinical trial	Submaximal cardiopulmonary exercise performance improved after six months of TSH normalization	23 patients with subclinical hypothyroidism, 11 treated and 12 untreated

**Table 2 biomedicines-13-02352-t002:** Main information about the scientific study and the key findings on the effects of physical activity and sport on the course of autoimmune thyroid disease.

Publication	Type of Publication	Main Findings	Study Group
Tian et al., (2024) [38]	Cross-sectional	Thyroid function is strongly affected by higher weekly physical activity and physical activity time, and there is a non-linear relationship between physical activity and thyroid disease. Patients with autoimmune thyroid disease maintained adequate levels of physical activity	Data from The National Health and Nutrition Examination Survey (NHANES) 2007–2012, 5877 American adults
Vuletić et al., (2024) [39]	Clinical trial	Unlike recreational exercise, occupational physical activity correlates with decreased thyroid function and increased thyroid autoimmunity in patient with overt hypothyroidism	438 individuals with clinically overt and subclinical hypothyroidism
Almas et al., (2023) [40]	Clinical trial	12 weeks of endurance training improve HR on-kinetics and physical activity level in subclinical hypothyroidism	18 women with subclinical hypothyroidism
Matsumura et al., (2015) [41]	Clinical trial	Training intensity and duration, including average miles per week, training pace, or years of accumulated running were not associated with thyroid dysfunction.	1222 female nonelite runners aged ≥ 35 years
Duñabeitia et al., (2009) [42]	Systematic review	Exercise showed a non-significant trend towards reducing thyroid-stimulating hormone levels	10 studies, 120 patients

## Data Availability

The data used to support the findings are fully available within the article itself.

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
