# Peer review of "Autoimmune Thyroid Diseases and Physical Activity and Sports—More Unknowns than Facts"

_biomedicines, 2025, doi:10.3390/biomedicines13102352_

Round 1
Reviewer 1 Report
Comments and Suggestions for Authors
This review aimed to determine which type and intensity of physical activity has the best-documented impact on autoimmune thyroid diseases, to quantify changes in thyroid function associated with physical activity, and to investigate the underlying mechanisms and influencing factors. The medical database PubMed/MEDLINE was searched from 2004 to 2025, and 12 publications met the inclusion criteria, ultimately being included in further evaluation according to the RAMESES (Realist and Meta-narrative Evidence Syntheses: Evolving Standards) framework. Results. The reviewed studies demonstrated that physical activity has a beneficial effect on thyroid function, and two studies reported that excessive physical activity leads to a decrease in TPO-Ab concentrations. Although all reviewed studies have shown that moderate physical activity has a beneficial effect on thyroid function and immune processes, there is insufficient evidence to formulate specific recommendations regarding the type and extent of physical activity for patients with autoimmune thyroid disease, nor to establish clear guidelines for participation in sports. The authors should include a brief addition to the introduction regarding the mentioned pathophysiological mechanisms of reduced exercise capacity in HT.
Author Response
Response to Review 1
We greatly appreciate your time and effort dedicated to providing feedback on our manuscript and we are grateful for the insightful comments on and valuable improvements to our paper. All the suggestions helped us to evaluate our outcomes even more precisely in order to deliver improved, high quality scientific manuscript which we hope will now meet the high standards of Biomedicines.
This review aimed to determine which type and intensity of physical activity has the best-documented impact on autoimmune thyroid diseases, to quantify changes in thyroid function associated with physical activity, and to investigate the underlying mechanisms and influencing factors. The medical database PubMed/MEDLINE was searched from 2004 to 2025, and 12 publications met the inclusion criteria, ultimately being included in further evaluation according to the RAMESES (Realist and Meta-narrative Evidence Syntheses: Evolving Standards) framework. Results. The reviewed studies demonstrated that physical activity has a beneficial effect on thyroid function, and two studies reported that excessive physical activity leads to a decrease in TPO-Ab concentrations. Although all reviewed studies have shown that moderate physical activity has a beneficial effect on thyroid function and immune processes, there is insufficient evidence to formulate specific recommendations regarding the type and extent of physical activity for patients with autoimmune thyroid disease, nor to establish clear guidelines for participation in sports.
The authors should include a brief addition to the introduction regarding the mentioned pathophysiological mechanisms of reduced exercise capacity in HT.
Response: We appreciate the Reviewer’s comment. Following the Reviewer's suggestion, a paragraph on the pathophysiological mechanisms of reduced exercise capacity in HT has been added to the Introduction section on page 2-3, lines: 89-114.
“Exercise capacity is substantially and directly impacted by thyroid hormones' influence on endurance, muscle strength and post-exercise regeneration. The positive effect of thyroid hormones on endurance is complex. They exert a permissive effect on catecholamines, increasing the expression of beta receptors, which results in increased heart rate, stroke volume, cardiac efficiency, and contractility [20]. In the respiratory system, they increase oxygenation [20]. Thyroid hormones stimulate mitochondriogenesis and thereby augment cellular oxidative capacity [21]. They stimulate mitochondrial oxidative phosphorylation, which leads to ATP production [21]. They also improve endurance by regulating enzymes involved in both lipolysis and lipogenesis as well as insulin-dependent glucose uptake, and both gluconeogenesis and glycogenolysis, leading to increased supply of energy substrates during prolonged physical exercise [22, 23]. Thyroid hormone deficiency has the opposite effect on all of the above-described processes and leads to reduced endurance. Thyroid hormones also produce a complex effect on skeletal muscles. They regulate the growth and differentiation of fast-twitch type II muscle fibres that are necessary for powerful movements [23]. The impact of thyroid hormones on skeletal muscle properties includes: increased rate of contraction and relaxation, decreased energetic efficiency of contraction due to higher ATP consumption at rest and during activity, increased glycolytic capacity and increased mitochondrial density which both lead to enhanced ATP generation [24]. Therefore, thyroid hormone deficiency may impair muscle strength, contraction and relaxation. Moreover, thyroid hormones support muscle remodelling and repair after intense exercise. They help reduce muscle discomfort and accelerate recovery by stimulating protein synthesis and improving the turnover of damaged proteins [25]. In hypothyroidism associated with Hashimoto's disease, the post-exercise regeneration process is weakened.”
The inserted excerpt entailed the addition of new references and a change in the numbering of subsequent references:
20 Kim B, Carvalho-Bianco SD, Larsen PR. Thyroid hormone and adrenergic signalling in the heart. Arq Bras Endocrinol Metabol. 2004 Feb;48(1):171-5. doi: 10.1590/s0004-27302004000100019.
21 Harper ME, Seifert EL. Thyroid hormone effects on mitochondrial energetics. Thyroid. 2008 Feb;18(2):145-56. doi: 10.1089/thy.2007.0250
22 Mullur R, Liu YY, Brent GA. Thyroid hormone regulation of metabolism. Physiol Rev. 2014 Apr;94(2):355-82. doi: 10.1152/physrev.00030.2013.
23 Cicatiello AG, Di Girolamo D, Dentice M. Metabolic Effects of the Intracellular Regulation of Thyroid Hormone: Old Players, New Concepts. Front Endocrinol (Lausanne). 2018 Sep 11;9:474. doi: 10.3389/fendo.2018.00474.
24 Salvatore D, Simonides WS, Dentice M, Zavacki AM, Larsen PR. Thyroid hormones and skeletal muscle--new insights and potential implications. Nat Rev Endocrinol. 2014 Apr;10(4):206-14. doi: 10.1038/nrendo.2013.238.
25 Nappi A, Moriello C, Morgante M, Fusco F, Crocetto F, Miro C. Effects of thyroid hormones in skeletal muscle protein turnover. J Basic Clin Physiol Pharmacol. 2024 Sep 20;35(4-5):253-264. doi: 10.1515/jbcpp-2024-0139.
Following the Reviewer's suggestion, the article was linguistically corrected.

Reviewer 2 Report
Comments and Suggestions for Authors
The scope of the paper is a known issue, and I believe it should highlight some novelty in the findings of the study. As a systematic review, it is expected to present some novel ideas; otherwise, the association between exercise and thyroid autoimmune disorders and related inflammation is an obvious known matter. Sport is immune suppressing and consequently autoimmunity, but what this paper can present should be highlighted in the abstract and text.
- A graphical abstract is highly demanded.
- The keyword should be corrected to be more connected and specific to the topic: sport is not a good keyword for this paper.
- The review includes only 12 studies, which is a small number over a 21-year window (2004–2025). This sparse dataset limits the comprehensiveness and robustness of the synthesized conclusions.
- Including a flow chart of study selection (PRISMA or similar) would improve transparency.
- The review does not attempt any meta-analysis or quantitative pooling of data, which is understandable given heterogeneity but limits objectivity. The presentation is predominantly narrative, relying on summary tables without effect size measures or statistical assessments.
- The manuscript currently lacks figures such as: Study selection flowchart showing screening and exclusion, Conceptual model diagrams illustrating key mechanisms or hypothetical pathways, Graphical summaries or evidence maps. Potential confounding factors (e.g., age, sex, disease duration, treatment status) impacting physical activity or thyroid function.
- The discussion on thyroid hormone use as doping is interesting, but could be better related to patient management or sports medicine contexts.
It should be improved
Author Response
Response to Review 2
Thank you for the time and effort dedicated to the review of our manuscript, and for your helpful comments and suggestions. We are particularly grateful for the Reviewer's comments and suggestions regarding the graphical representation of the conceptual process, methodology, and results, and for suggesting specific visualization tools. Their use makes the work clearer and more understandable.
Below we have included responses to the Reviewer's specific comments.
The scope of the paper is a known issue, and I believe it should highlight some novelty in the findings of the study. As a systematic review, it is expected to present some novel ideas; otherwise, the association between exercise and thyroid autoimmune disorders and related inflammation is an obvious known matter. Sport is immune suppressing and consequently autoimmunity, but what this paper can present should be highlighted in the abstract and text.
- The scope of the paper is a known issue” and „the association between exercise and thyroid autoimmune disorders and related inflammation is an obvious known matter.”
Response:
Indeed, autoimmune thyroid diseases are very common and the mechanisms by which thyroid hormones affect energy processes and the muscular system are well known. We also know the main symptoms of thyroid disease, including a reduced ability to engage in physical activity. Yet, despite theoretical premises, our analysis shows that there are no reliable clinical or even survey-based studies assessing this issue. Physical exercise has been reported to have a positive impact on the immune system. However, our paper shows that almost no research has been conducted for autoimmune thyroid diseases. No recommendations regarding physical activity in autoimmune thyroid diseases are available either. There is no single publication regarding the return to sports during the treatment of hyperthyroidism in Graves' disease. The novelty of our work lies in its comprehensive summary of a previously unexplored topic and in the identification of the effects of exercise of varying intensity on immunological thyroid diseases and the recommendations for its implementation.
- „As a systematic review, it is expected to present some novel ideas”.
Response:
We would like to thank the Reviewer for pointing this out because this made us realize that in the methodology section, we had not included a sufficient discussion of the reasons for the methodology selection. Due to the nature of the problem addressed and the desire to determine whether recommendations can be made based on existing research or whether the creation of specific research programs is required, the authors utilized a realist meta-narrative review. This type of review is a theory-driven and interpretive type of literature review which aims to answer the questions whether some
type of intervention is effective, if so, for whom, in what circumstances, and how. It is especially useful with respect to social intervention programs (Pawson et al., 2004). This type of review lets
„Articulate underlying programme theories and then to interrogate the existing evidence to find out whether and where these theories are pertinent and productive” (Pawson et al., 2006), and we trust that our investigation meets these assumptions of a realistic meta-narrative review.
To address this problem, the "Methods" section was changed as follows (page 4, lines 148-160):
„This review was conducted in accordance with RAMESES (Realist and Meta-narrative Evidence Syntheses: Evolving Standards) guidelines, following the principles of a realist review [29,30]. This method was selected because it is a theoretically and interpretively based literature review that seeks to answer the question of whether a given intervention works, for whom, under what circumstances, and how. It enables the development of basic theories for an intervention program and subsequent analysis of the existing evidence to assess their validity and practical applicability. The aim of this review was to identify the type and intensity of physical activity with the most well-documented influence on autoimmune thyroid disease, to quantify changes in thyroid function associated with physical activity, to explore the underlying mechanisms and influencing factors and present theory for intervention program aimed to formulate recommendations.
Conceptual model diagrams illustrating key mechanisms or hypothetical pathways are presented in Figure 1.”
- Sport is immune suppressing and consequently autoimmunity, but what this paper can present should be highlighted in the abstract and text.
Response: We thank the Reviewer for pointing this out. The following paragraph has been added to the introduction (page 3 lines 122-127).
„On the other hand, intense physical exercise leads to a transient immune dysfunction. Specifically, for several hours or days after prolonged and intense physical exercise, T-cell, NK, and neutrophil function is impaired, the balance of type I and II cytokines is disturbed, the expression of major histocompatibility complex II in macrophages is altered. Additionally, the immune responses to primary and recall antigens are weakened, and delayed [28]. “
The word “non-excessive” has also been added to the sentence on page 3, line 115:
“On the other hand, non-excessive physical activity has been shown to significantly increase the number of T-regulatory cells, shift the Th1/Th2 balance towards a reduced Th1 cell production and decreased immunoglobulin secretion [26,27].”
The word “moderate” has been added to the sentence on page 3, line 129:
“Although pathophysiological mechanisms suggest impaired exercise capacity in both HD and GD, and there is interest in the beneficial benefits of moderate physical activity on the immunological processes underlying these diseases, there are, to the authors' knowledge, no clear recommendations regarding activity and sports participation in individuals suffering from them.”
The deifications entailed the addition of one new reference and a change in the numbering of subsequent references:
- Nieman DC, Wentz LM. The compelling link between physical activity and the body's defence system. J Sport Health Sci. 2019 May;8(3):201-217. doi: 10.1016/j.jshs.2018.09.009
The issue pointed out by the Reviewer was also indicated in the abstract which was complemented with the following sentence:
„In turn, it is known that moderate physical exercise can have a positive effect on the immune system, while excessive exercise can cause unfavourable changes in this system.”
- A graphical abstract is highly demanded.
Response: We would like to thank the Reviewer for this remark. As suggested by the Reviewer, the graphical abstract was included during the resubmission of the manuscript.
- The keyword should be corrected to be more connected and specific to the topic: sport is not a good keyword for this paper.
Response: We thank the Reviewer for pointing this out. As suggested, "sport" was removed from the keywords.
- The review includes only 12 studies, which is a small number over a 21-year window (2004–2025). This sparse dataset limits the comprehensiveness and robustness of the synthesized conclusions.
Response: We would like to thank the Reviewer again for raising this issue.
Indeed, over the past 20 years, the number of studies on our topic has been surprisingly limited. This prompted the authors to identify the type and intensity of physical activity with the most well-documented impact on autoimmune thyroid disease, to quantify the changes in thyroid function associated with physical activity, to explore the underlying mechanisms and factors influencing these changes, and to present a theory of an intervention program with a view to formulating recommendations. To this end, as explained the above, we employed a realist and meta-narrative evidence syntheses as the most appropriate method. To address this problem, the "Methods" section was modified as discussed above.
- Including a flow chart of study selection (PRISMA or similar) would improve transparency.
Response: We thank the Reviewer for this comment. Our Figures 1 and 2 did not show screening and exclusion. Therefore, they were replaced with Figure 2 and Figure 3, which is a detailed flow chart based on the RAMESES (Realist and Meta-narrative Evidence Syntheses: Evolving Standards) guidelines. (Page 7, line 211-218).
The captions of the figures have been changed (page 7, lines 213-216 and 219-221):
„Figure 2. Study selection flow chart for realist and meta-narrative evidence synthesis based on Realist and Meta-narrative Evidence Syntheses: Evolving Standards guidelines by Pawson R et al. [29]. Search terms: “physical activity”, “sports”, “autoimmune thyroid disease”, “autoimmune thyroiditis”, “Hashimoto's thyroiditis”, “Graves’s disease”, “hypothyroidism”, “hyperthyroidism”, and “thyrotoxicosis”. „
„Figure 3. Study selection flow chart for realist and meta-narrative evidence synthesis based on Realist and Meta-narrative Evidence Syntheses: Evolving Standards guidelines by Pawson R et al. [29]. Search terms: “thyroid hormones”, “levothyroxine, “triiodothyronine” and “doping”.”
- The review does not attempt any meta-analysis or quantitative pooling of data, which is understandable given heterogeneity but limits objectivity. The presentation is predominantly narrative, relying on summary tables without effect size measures or statistical assessments.
Response: Thank you for this comment. As we mentioned above, Realist and Meta-narrative Evidence Synthesis is a theory-driven and interpretive type of literature review with a meta-narrative character that does not focus on summative analysis. This type of review is the opposite of a systematic review, in which all empirical evidence meeting pre-defined criteria is compiled to answer a specific research question. The use of realist and meta-narrative evidence synthesis explains why “the presentation is predominantly narrative, relying on summary tables without effect size measures or statistical assessments”.
- The manuscript currently lacks figures such as:
-Study selection flowchart showing screening and exclusion,
-Conceptual model diagrams illustrating key mechanisms or hypothetical pathways,
-Graphical summaries or evidence maps
Response: We wish to express our special thanks to the Reviewer for pointing this out. As suggested, the publication now includes: a study selection flowchart showing screening and exclusion (page 7, lines 211-218) and their captions (page 7, lines 213-217 and 219-221), conceptual model diagrams illustrating key mechanisms or hypothetical pathways (page 5, lines 198-199) and its caption (page 5, lines 200-201), and evidence gap maps of findings (page 9, lines 239-244) and their captions (page 9 lines 237-238, 241-243).
- Potential confounding factors (e.g., age, sex, disease duration, treatment status) impacting physical activity or thyroid function.
Response: Thank you for this comment. The following paragraph has been included in the limitations of the study (page 14, lines 472-478):
“Another limitation is that the analysed publications did not provide sufficient evidence to determine whether factors such as gender, age, or disease duration modified the effect of exercise on the immune system and thyroid function in autoimmune disease. Similarly, only a limited number of studies accounted for treatment as a potential confounding factor. While some findings suggested that the impact of exercise on thyroid function may be more pronounced in treated patients, the available data are insufficient to support definitive conclusions.”
- The discussion on thyroid hormone use as doping is interesting, but could be better related to patient management or sports medicine contexts.
Response: We thank the Reviewer for this comment. In response, the first paragraph for 4.3 section was amended as follows: (page 12, lines 366-381)
“Although thyroid hormones are not classified as doping agents by the World Anti-Doping Agency (WADA), there are theoretical grounds that may explain the possible influence of thyroid hormones on the ability to improve performance. They increase the expression of beta receptors, exerting a permissive effect on catecholamines. [20]. They stimulate mitochondriogenesis, which results in increased cellular oxidative capacity and mitochondrial oxidative phosphorylation, increasing ATP production [21]. They also regulate enzymes involved in lipolysis, lipogenesis, insulin-dependent glucose uptake, and both gluconeogenesis and glycogenolysis, thereby increasing the supply of energy substrates. [23, 24]. They regulate the growth and differentiation of fast-twitch type II muscle fibres, increase the rate of contraction and relaxation, reduce the energy efficiency of contraction due to higher ATP consumption at rest and during activity, increase glycolytic capacity with subsequent ATP generation, and increase mitochondrial density, leading to increased ATP generation [24, 25]. Exogenously administered TH are therefore attractive to professional athletes and bodybuilders who want to achieve an appropriate weight and low body fat. Additionally, these drugs are illegally used to counteract changes in thyroid hormone levels that can occur during the use of anabolic-androgenic steroids [28].”
Additionally, we have added two new paragraphs on pages 12, lines 402-411 and page 13 lines 450-461):
“Testosterone, in turn, was used illegally by 35% of the surveyed amateur athletes. Although thyroid hormones were used by only about 12% of those who trained, the potential for serious complications resulting from their use may pose a serious toxicological problem. To the authors' knowledge, there are no available studies summarizing the toxicological significance of illegal thyroid hormone use, however, numerous clinical case reports document instances of poisoning in bodybuilding athletes, including a fatal case [49-58]. From a clinical management perspective, it is essential to consider the possibility of illicit drug use in individuals practicing sports such as bodybuilding when analysing laboratory tests and clinical presentations. Early detection is critical to avoid misdiagnosis or delayed treatment. [59]. Furthermore, both primary and secondary prevention strategies should also be implemented to mitigate thyroid hormones abuse.”
“From a sports medicine perspective, the inclusion of thyroid hormones on the list of prohibited substances in sport will necessitate the use of Therapeutic Use Exemption (TUE), which means that athletes would have to obtain WADA approval to use the medication for medically justified reasons. Given the prevalence of Hashimoto's thyroiditis, which requires thyroid hormone treatment, such a policy change could significantly increase the workload on the regulatory bodies responsible for evaluating TUE applications and may complicate the approval process. Importantly, without a valid TUE, athletes undergoing thyroid hormone therapy would be ineligible to participate in competitive sports.”
This involved adding 11 new citations:
- Gild ML, Stuart M, Clifton-Bligh RJ, Kinahan A, Handelsman DJ. Thyroid Hormone Abuse in Elite Sports: The Regulatory Challenge. J Clin Endocrinol Metab. 2022 Aug 18;107(9):e3562-e3573. doi: 10.1210/clinem/dgac223.
- Mark PB, Watkins S, Dargie HJ. Cardiomyopathy induced by performance enhancing drugs in a competitive bodybuilder. Heart. 2005 Jul;91(7):888. doi: 10.1136/hrt.2004.053843.
- Chen YC, Fang JT, Chang CT, Chou HH. Thyrotoxic periodic paralysis in a patient abusing thyroxine for weight reduction. Ren Fail. 2001 Jan;23(1):139-42. doi: 10.1081/jdi-100001294
- Kwak T, Al Zoubi M, Bhavith A, Rueda Rios C, Kumar S. Acute myocarditis in bodybuilder from coxsackievirus and thyrotoxicosis. J Cardiol Cases. 2016 Jul 20;14(4):123-126. doi: 10.1016/j.jccase.2016.06.005.
- Daher G, Hassanieh I, Malhotra N, Alderson L. Acute Decompensated Heart Failure Secondary to Exogenous Triiodothyronine Use in a Young Non-athlete Weightlifter. Cureus. 2019 Oct 22;11(10):e5964. doi: 10.7759/cureus.5964.
- Roomi S, Ullah W, Iqbal I, Ahmad A, Saleem S, Sattar Z. Thyrotoxicosis factitia: a rare cause of junctional rhythm and cardiac arrest. J Community Hosp Intern Med Perspect. 2019 Jun 19;9(3):258-263. doi: 10.1080/20009666.2019.1618668
- Patel AJ, Tejera S, Klek SP, Rothberger GD. THYROTOXIC PERIODIC PARALYSIS IN A COMPETITIVE BODYBUILDER WITH THYROTOXICOSIS FACTITIA. AACE Clin Case Rep. 2020 Sep 21;6(5):e252-e256. doi: 10.4158/ACCR-2020-0154
- van Bokhorst QNE, Krul-Poel YHM, Smit DL, de Ronde W. A 29-year-old Bodybuilder with Liothyronine-induced Thyrotoxic Hypokalaemic Periodic Paralysis. Eur J Case Rep Intern Med. 2021 Mar 4;8(3):002362. doi: 10.12890/2021_002362
- Bonnar CE, Brazil JF, Okiro JO, Giblin L, Smyth Y, O'Shea PM, Finucane FM. Making weight: acute muscle weakness and hypokalaemia exacerbated by thyrotoxicosis factitia in a bodybuilder. Endocrinol Diabetes Metab Case Rep. 2021 Oct 1;2021:21-0060. doi: 10.1530/EDM-21-0060
- Momoh R, Hassan A. A Case Report of an Acute Severe Tachyarrhythmia Presentation With Underlying Cardiomyopathy in a Patient With Anabolic Androgenic Steroid and Thyroxine Misuse. Cureus. 2024 Jun 21;16(6):e62806. doi: 10.7759/cureus.62806
- Warner BE, Woodrow CJ, Pal A. Delayed diagnosis of T3 supplementation in a bodybuilder presenting with tachycardia and features of sepsis. BMJ Case Rep. 2020 Jan 13;13(1):e232867. doi: 10.1136/bcr-2019-232867
- Comments on the Quality of English Language
It should be improved
Response: Following the Reviewer's suggestion, the article was linguistically corrected.

Round 2
Reviewer 2 Report
Comments and Suggestions for Authors
The authors revised the paper perfectly, and I have no more comments.